# RuO$_2$ electronic structure and lattice strain dual engineering for enhanced acidic oxygen evolution reaction performance

Yin Qin[1,10], Tingting Yu[1,10], Sihao Deng[2], Xiao-Ye Zhou [3✉], Dongmei Lin[4], Qian Zhang[5], Zeyu Jin[1], Danfeng Zhang [6], Yan-Bing He[6], Hua-Jun Qiu [1✉], Lunhua He[2,7,8], Feiyu Kang[6], Kaikai Li [1✉] & Tong-Yi Zhang [9✉]

Developing highly active and durable electrocatalysts for acidic oxygen evolution reaction remains a great challenge due to the sluggish kinetics of the four-electron transfer reaction and severe catalyst dissolution. Here we report an electrochemical lithium intercalation method to improve both the activity and stability of RuO$_2$ for acidic oxygen evolution reaction. The lithium intercalates into the lattice interstices of RuO$_2$, donates electrons and distorts the local structure. Therefore, the Ru valence state is lowered with formation of stable Li-O-Ru local structure, and the Ru–O covalency is weakened, which suppresses the dissolution of Ru, resulting in greatly enhanced durability. Meanwhile, the inherent lattice strain results in the surface structural distortion of Li$_x$RuO$_2$ and activates the dangling O atom near the Ru active site as a proton acceptor, which stabilizes the OOH* and dramatically enhances the activity. This work provides an effective strategy to develop highly efficient catalyst towards water splitting.

[1] School of Materials Science and Engineering, Harbin Institute of Technology (Shenzhen), 518055 Shenzhen, China. [2] Spallation Neutron Source Science Center, 523803 Dongguan, China. [3] School of Civil Engineering, Shenzhen University, 518060 Shenzhen, Guangdong, China. [4] Department of Mechanical Engineering, Research Institute for Smart Energy, The Hong Kong Polytechnic University, Hong Kong SAR, China. [5] Materials Genome Institute, Shanghai University, 333 Nanchen Road, 200444 Shanghai, China. [6] Shenzhen All-Solid-State Lithium Battery Electrolyte Engineering Research Center, Institute of Materials Research (IMR) Tsinghua Shenzhen International Graduate School, Tsinghua University Shenzhen, 518055 Shenzhen, China. [7] Beijing National Laboratory for Condensed Matter Physics, Institute of Physics, Chinese Academic of Sciences, 100190 Beijing, China. [8] Songshan Lake Materials Laboratory, 523808 Dongguan, China. [9] The Hong Kong University of Science and Technology (Guangzhou), Advanced Materials Thrust and Sustainable Energy and Environment Thrust, Nansha, Guangzhou, 511400 Guangdong, China. [10] These authors contributed equally: Yin Qin, Tingting Yu. ✉email: xiaoye_zhou@szu.edu.cn; qiuhuajun@hit.edu.cn; likaikai@hit.edu.cn; zhangty@shu.edu.cn

The oxygen evolution reaction (OER) is a crucial anodic reaction in electrochemical water splitting[1–4]. Intrinsically, the process of OER involves a four-electron transference, which demands higher energy than the cathodic reaction, i.e., hydrogen evolution reaction (HER) which needs only two electrons[5,6]. Therefore, the OER process governs the overall efficiency of electricity-driven water splitting. Water splitting can be operated in either acidic or alkaline conditions. OER under acidic conditions are more preferable benefiting from the higher ionic conductivity of acidic electrolyte and capability of operating at higher current density as well as more compact system design[7–9], but their practical application is significantly hindered by the sluggish OER kinetics and limited stability of existing electrocatalysts[10–12]. Thus, it is imperative to develop acidic OER electrocatalysts with enhanced activity and stability in order to improve the efficiency of electrochemical water splitting.

Rutile $RuO_2$ is considered as a benchmark catalyst for the acidic OER[13]. Nevertheless, the low activity of virgin $RuO_2$ and the poor stability as a result of the dissolution of Ru and participation of lattice oxygen (lattice oxygen-mediated mechanism, LOM) in acidic media remain serious problems for $RuO_2$ catalysts[14–16]. In order to improve the performance of $RuO_2$ electrocatalysts, tuning the electronic structure of Ru sites by lattice doping has been demonstrated to be an effective strategy[11,17–20]. In particular, first-row transition metals are usually considered as doping elements owing to their unique features of $3d$ electrons and low cost[4,16,18,21,22]. Other transition metals such as Y[19], Pt[11], W, and Er[23] were also reported as effective doping elements. The charge density and spin density of $RuO_2$ can be redistributed by doping with these alien atoms of different valence state and electronegativity, thus regulating the adsorption energy of the oxo-intermediates at active sites[12,17,18,24]. The doped $RuO_2$, e.g., Co-doped $RuO_2$[25], may follow a LOM mechanism because of the increase of the covalency of the metal–oxygen bonds[26], rather than the conventional adsorbate evolution mechanism (AEM), resulting in enhanced activity but probably poor stability due to the oxidation of lattice oxygen. Although W, Er- co-doping strategy was reported to be able to enhance the energy barrier of the lattice oxygen oxidation of $RuO_2$ and prohibit the formation of oxygen vacancies due to the enlarged gap between the Fermi level and the O $2p$-band center[23], there is still much room to enhance the stability and activity of $RuO_2$ for practical applications.

In addition to doping, electrochemical ion insertion involving coupled ion–electron transfer is also an effective method to introduce alien elements into a host material for electronic or crystal structure modulation, and has been considered as a synthetic strategy to improve the catalytic performance of layer-structured materials[27–29], such as $LiCoO_2$ for OER[30] and $MoS_2$ for HER[28], where the Li concentration is an adjustable variable over a wide range[31,32]. Recently, Zheng's group utilized a lithiation strategy to improve the $CO_2$ reduction performance of catalysts, including $Cu_3N_x$[33] and Sn[34]. Various studies have shown that $RuO_2$ can be inserted with Li ions for battery applications, and a solid solution phase forms before a Li:Ru = 1:1 ratio is reached[35–38]. On the other hand, the insertion of a large amount of lithium atoms into $RuO_2$ may induce a relatively large lattice strain. Nevertheless, engineering lattice strain by electrochemical lithium insertion has not been fully explored for improving OER performance of $RuO_2$.

In this work, we adopt an electrochemical method to intercalate lithium into $RuO_2$ lattice interstices with tunable lithium concentration to improve the OER activity and durability of $RuO_2$ in acidic media. We find that the OER activity of the formed $Li_xRuO_2$ solid solution phase increases with the nominal lithium concentration ($x$) and reaches a record low overpotential

of 156 mV at 10 mA $cm^{-2}$ in 0.5 M $H_2SO_4$ when $x$ reaches 0.52. Meanwhile, the $Li_{0.52}RuO_2$ exhibits excellent durability during 70 h chronopotentiometry test with neglectable overpotential increase. XAS analysis and DFT calculations reveal that lithium, as an electron donor, influences the electronic structure and lattice strain of $RuO_2$. The Ru−O $4d − 2p$ hybridization is weakened with a decreased Ru–O covalency. Meanwhile, the valence state of Ru is decreased with the formation of stable Li–O-Ru local structure. Thus, the participation of lattice oxygen and dissolution of Ru are suppressed during OER, enhancing the stability of $RuO_2$. DFT calculations find that the surface structural distortion induced by inherent lattice strain activates the dangling O atom near the Ru active site as a proton acceptor to stabilize the OOH* and thus dramatically enhances the activity of $RuO_2$. This work proposes a creative strategy to design highly efficient and stable OER catalysts.

## Results and discussion

**Crystal structure and composition.** Lithium intercalated $RuO_2$ ($Li_xRuO_2$) with tunable lithium concentration was prepared by electrochemical lithiation process which involves coupled ion–electron transfer, as shown in Fig. 1a. The lithium concentration $x$ in $Li_xRuO_2$ is linearly correlated to the time when the current density is constant during electrochemical lithiation, and thus can be easily adjusted. Rutile $RuO_2$ crystallizes in a tetragonal system with a space group of $P4_2/mnm$, consisting of a ruthenium atom octahedrally coordinated to six oxygen atoms (Fig. 1b)[39]. *Operando* XRD, ex situ XRD, and TEM were conducted to reveal the crystal structure of the $Li_xRuO_2$ after lithium intercalation. The *operando* XRD (Fig. 1d and Supplementary Fig. 1) results under a constant current density of 10 mA $g^{-1}$ indicate that a solid solution phase $Li_xRuO_2$ with the same rutile structure to pristine $RuO_2$ formed in the initial stage of lithium intercalation, evidenced by the slight shifting of the original peaks towards the lower angles. Further lithiation induces a first-order phase transition from the solid solution phase to $LiRuO_2$ phase[32], as a new set of diffraction peaks appears. However, the $LiRuO_2$ phase is unstable when the electrochemical lithiation process is terminated. The intensity of the XRD peaks of the $LiRuO_2$ phase gradually weakens while the peaks of $Li_xRuO_2$ phase strengthen during relaxation, indicating the reversed transition from $LiRuO_2$ to $Li_xRuO_2$ phase. Thus, the final structure of the $RuO_2$ after lithium intercalation is $Li_xRuO_2$, a solid solution phase, which is further confirmed by the ex situ TEM and XRD results (Fig. 1e, f). Figure 1e presents the ex situ XRD patterns of the pristine $RuO_2$ and the $Li_xRuO_2$ after electrochemical lithium intercalation under a current density of 10 mA $g^{-1}$ for 2 h, 9 h, 12 h, and 16 h, corresponding to the nominal lithium concentrations of $x = 0.07$, 0.29, 0.39, and 0.52 (Details for the estimation of the nominal lithium concentration can be found in Supplementary Fig. 2). Obviously, the $Li_xRuO_2$ inherits the XRD characteristics of the pristine $RuO_2$ with the shift of XRD peaks towards low angles (Supplementary Fig. 3), which means the lattice of the $RuO_2$ was expanded due to lithium intercalation. Neutron powder diffraction (NPD) analyses (Supplementary Fig. 4) and DFT calculations (Supplementary Figs. 5 and 6 and Fig. 1c) indicate that the lithium ions intercalate into the octahedral interstice formed by six adjacent O atoms rather than replacing the Ru cations, and thereby the $RuO_2$ lattice is expanded, which is in line with the XRD results. To extract the lattice parameters of the $RuO_2$ before and after lithium intercalation, an Expectation–Maximization (EM) Algorithm-based machine-learning method was adopted to fit the XRD patterns. The fitting results are illustrated in Supplementary Fig. 7, and the lattice parameters of all the samples are listed in Supplementary Table 1. A dilatation strain along the

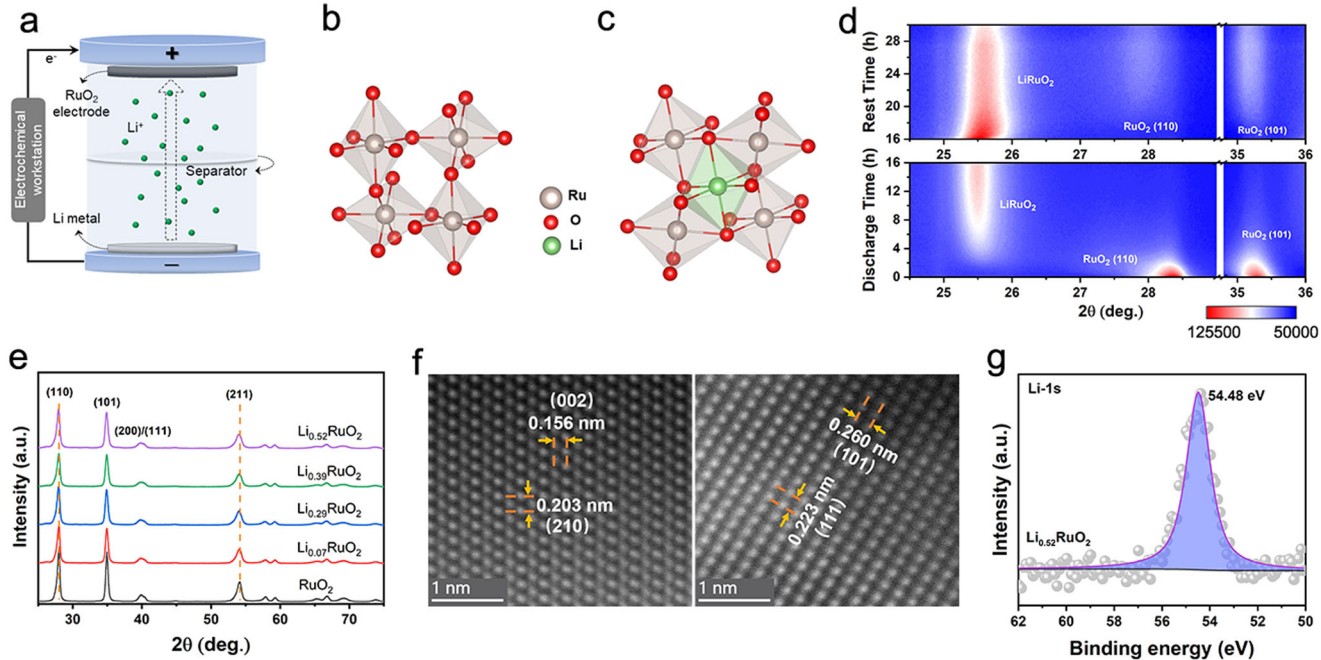

**Fig. 1 Structural and compositional characterizations. a** Schematic illustration of the preparation of lithium intercalated $RuO_2$. **b** $RuO_6$ octahedron before lithium intercalation. **c** $RuO_6$ octahedron after lithium intercalation. **d** *Operando* XRD of $RuO_2$ during electrochemical lithiation under a constant current density of 10 mA g$^{-1}$, followed by 14 h relaxation. **e** Ex situ XRD patterns of the pristine $RuO_2$ and the $Li_xRuO_2$. **f** The HAADF-STEM images of the pristine $RuO_2$ (left) and the $Li_{0.52}RuO_2$ (right). **g** The high-resolution Li 1*s* XPS of $Li_{0.52}RuO_2$.

*a*-axis is observed and increases from 0.14 to 0.25% with the increase of the degree of lithiation. The HAADF-STEM images show the lattice fringes corresponding to the (002), (210) planes (Fig. 1f, left), and (101), (111) planes (Fig. 1f, right) of rutile-structured $RuO_2$, further demonstrating that the $RuO_2$ after lithium intercalation preserves its original crystal structure. In addition, the lithium intercalation shows no influence on the morphology of the $RuO_2$ particles (Supplementary Fig. 8). In contrast to pristine $RuO_2$, the presence of a Li 1*s* peak in the X-ray photoelectron spectroscopy (XPS) profile of $Li_{0.52}RuO_2$ (Fig. 1g and Supplementary Fig. 9a) indicates that lithium is inserted. Furthermore, the Li K-edge (edge onset at 55 eV) STEM-EELS map (Supplementary Fig. 9b, c) and EDS elemental map (Supplementary Fig. 10) of the $Li_{0.52}RuO_2$ confirm the existence of lithium.

**Catalytic performance**. The OER performance of the pristine $RuO_2$ and $Li_xRuO_2$ was evaluated using a three-electrode system in 0.5 M $H_2SO_4$ solution. Fig. 2a shows the polarization curves measured by linear sweep voltammetry (LSV) with the current normalized by the disk area of the glassy carbon electrode. Supplementary Fig. 11 shows that the $O_2$ generation starts at around 1.3 V, and the polarization curve shows almost no change in the initial seven cycles. Here, the overpotential for reaching a current density of 10 mA cm$^{-2}$ ($\eta_{10}$) of the 3rd cycle is used for activity comparison. The pristine $RuO_2$ exhibits the lowest activity with an overpotential of 320 mV. As the lithium concentration $x$ increases, the overpotential gradually decreases and reaches a significantly low value of 156 mV for $Li_{0.52}RuO_2$ (Fig. 2b), which overcomes the limitation from the inherent linear scaling relation[44]. However, further increasing the lithium concentration does not make further improvement of the activity, and the $Li_{0.52}RuO_2$ exhibits the best activity. It is worth noting that the $Li_{0.52}RuO_2$ requires a small overpotential of 335 mV to deliver a large OER current density of 200 mA cm$^{-2}$. We further estimated

the electrochemically active surface area (ECSA) of $RuO_2$ and $Li_xRuO_2$, and plotted the LSVs with respect to the ECSA (Supplementary Figs. 12 and 13), which indicates that the higher OER activity of $Li_xRuO_2$ is not attributed to the varied ECSA, and the Li insertion plays an important role in enhancing the intrinsic activity. Tafel plots derived from the polarization curves within the overpotential range of 0.17 to 0.27 V, i.e., 1.4-1.5 V vs RHE, are shown in Fig. 2c. The Tafel slopes of the pristine $RuO_2$ and $Li_xRuO_2$ (where $x = 0, 0.07, 0.29, 0.39, 0.52$) are 105.8, 103.6, 87.7, 86.0, and 83.3 mV dec$^{-1}$, respectively. The decrease of Tafel slope with an increase in lithium concentration indicates that the electrocatalytic kinetics of $RuO_2$ are enhanced by lithium intercalation[2,14,45]. In addition, all the Tafel slopes are higher than 80 mV dec$^{-1}$, indicating that all the catalysts operate via the same OER mechanism[18,42,46].

In addition to activity, durability is another crucial parameter for evaluating the OER performance of electrocatalysts in acidic electrolyte due to the corrosive conditions. Chronopotentiometry tests were conducted at a current density of 10 mA cm$^{-2}$. As shown in Fig. 2d, the catalytic stability of $Li_{0.52}RuO_2$ is far better than that of the pristine $RuO_2$. The $Li_{0.52}RuO_2$ can continuously work for 70 h without an evident increase in the overpotential. In comparison, the OER activity of pristine $RuO_2$ decreases dramatically in less than 20 h. The dissolution of Ru in the acidic electrolyte during electrolysis is further monitored using inductively coupled plasma optical emission spectrometry (ICP-OES). The percentage of Ru dissolved from pristine $RuO_2$ and $Li_{0.52}RuO_2$ during the chronopotentiometry tests at 10 mA cm$^{-2}$ was measured and shown in Fig. 2e. For pristine $RuO_2$, the dissolution percentage of Ru is very low because of its low activity and poor stability. For $Li_{0.52}RuO_2$, in the 1st hour of the OER test, the dissolution percentage of Ru is around 0.9%. After 24 h, the dissolution percentage of Ru is increased slightly to 1.8%, and then plateaued. Even after 48 h, the dissolution percentage of Ru remained very low at 1.9% which is much lower than those reported for amorphous/crystalline hetero-phase $RuO_2$ (in 0.1 M

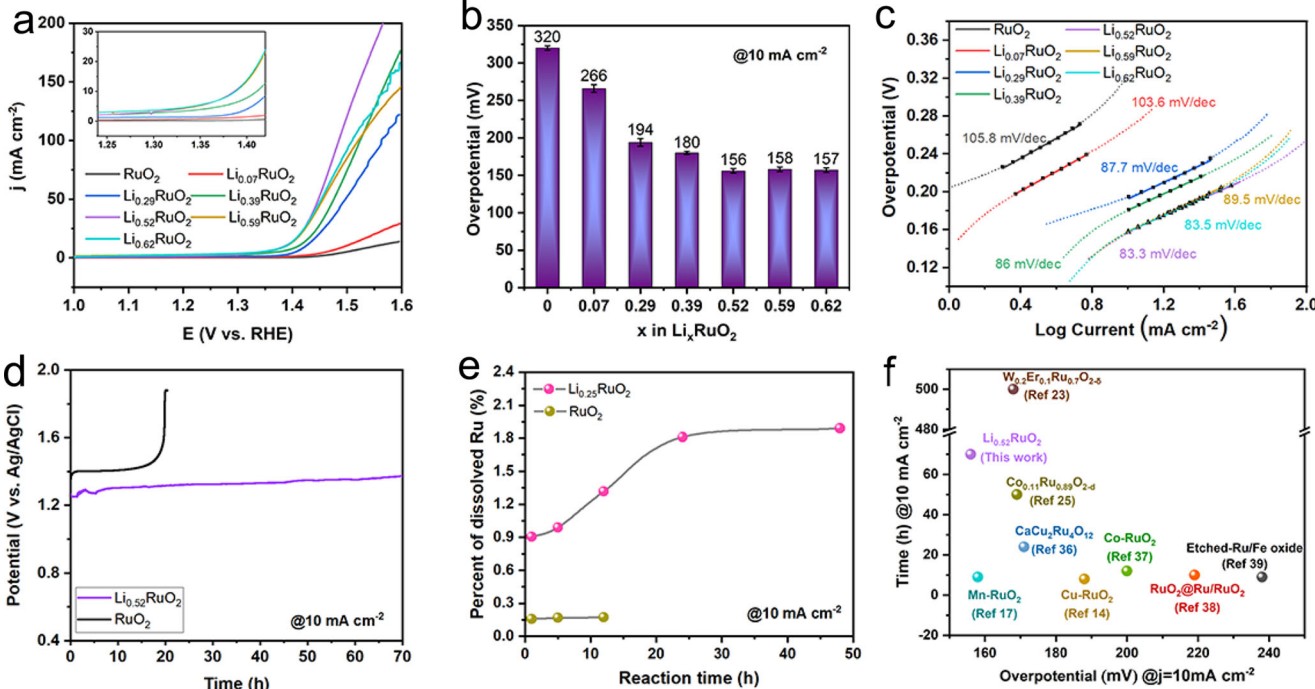

**Fig. 2 OER performance in 0.5 M $H_2SO_4$ solution. a** Polarization curves. RHE reversible hydrogen electrode. **b** Overpotentials ($\eta_{10}$) of $RuO_2$ and $Li_xRuO_2$ at 10 mA cm$^{-2}$. The error bars represent the deviation from the overpotentials in (**a**). **c** Tafel plots. **d** Chronopotentiometry curve of $Li_{0.52}RuO_2$ and $RuO_2$ at a current density of 10 mA cm$^{-2}$. **e** Percentage of Ru dissolved from $RuO_2$ and $Li_{0.52}RuO_2$ after electrocatalysis for different reaction times. **f** Comparison of the overpotential required to achieve a 10 mA cm$^{-2}$ cathodic current density and chronopotentiometry durability at 10 mA cm$^{-2}$ in acidic media for various $RuO_2$-based electrocatalysts[14,17,23,25,40–43].

$HClO_4$[47] and SrRuIr oxide (in 0.5 M $H_2SO_4$)[48] during chronopotentiometry test at 10 mA cm$^{-2}$, indicating good corrosion resistance of $Li_{0.52}RuO_2$ in acidic condition. In sum, the $Li_{0.52}RuO_2$ shows excellent activity and stability, outperforming many state-of-the-art $RuO_2$-based acidic OER electrocatalysts (Fig. 2f)[14,17,23,25,42,48].

**Origin of the enhanced activity**. The scaling relation among the OER intermediates in AEM imposes a theoretical overpotential ceiling on the OER activity[3], which is apparently overcome by $Li_{0.52}RuO_2$. To uncover the origin of the enhanced activity, density functional theory (DFT) calculations and X-ray absorption spectroscopy (XAS) analyses were performed to get insights into the electronic and crystal structures of the $Li_xRuO_2$. DFT calculations were performed on the superlattice of $Li_nRu_{32}O_{64}$ ($Li_xRuO_2$ with $x = n/32$) to reveal the influence of lithium intercalation on the electronic structure of $RuO_2$. The calculation results show that the $d$-band structure of Ru and $2p$-band structure of O are modulated (Fig. 3a) by lithium intercalation. The partial density of states (PDOS) analyses demonstrate that the $e_g$ occupancy is much closer to unity for $Li_{0.5}RuO_2$ ($|e_g - 1| = 0.05$) than $RuO_2$ ($|e_g - 1| = 0.16$), and meanwhile the O $2p$-band center moves closer to the Fermi level slightly. The $e_g$ occupancy is highly related to the binding strength of active Ru sites with oxo-intermediates, and the optimal OER activity is generally achieved when the $e_g$ occupancy is close to unity[3]. Thus, the activity enhancement by lithium intercalation is partially attributed to the modulation of the electronic structure of Ru.

However, only modulating the $e_g$ occupancy of Ru can hardly break the scaling relation for achieving better activity[49]. Activating the lattice O (LOM) by increasing the Ru–O covalency can avoid the limitation caused by the scaling relation, which is however demonstrated to be impossible for this case, as discussed

in the next section. Figure 3b shows the Fourier-transformed Ru K-edge extended X-ray absorption fine structure (EXAFS) spectra of pristine $RuO_2$ and $Li_xRuO_2$. All the spectra exhibit the same spectral components, but a slight loss in intensity and difference in peak position are observed with the increase in lithium concentration. The peaks represent the neighboring atomic shells in the vicinity of Ru, i.e., O in the first shell and Ru in the second shell. Fitting the Fourier-transformed EXAFS spectra determines the bond lengths and average coordination numbers. It is revealed that, as the lithium concentration is increased, the coordination number of Ru–O decreases slightly, which implies an intrinsically lattice distortion/strain induced by lithium intercalation and is in line with the broadening of the full width at half maximum (FWHM) of the XRD peaks (Supplementary Fig. 14) and the enhancement of the background intensity of NPD patterns (Supplementary Fig. 5). The lattice distortion/strain is also evidenced by HAADF-STEM. Figure 3c and Supplementary Fig. 15 show the lattice strain distributions of $RuO_2$ and $Li_xRuO_2$ calculated from geometric phase analysis (GPA) of atomic-resolution HAADF-STEM images and HRTEM images. Compared with the pristine $RuO_2$, the intercalation of lithium generates more intense tensile-compressing dislocation dipoles in these GPA strain maps due to the distortion of $RuO_2$ lattice. The stronger strain field of $Li_xRuO_2$ will give rise to a more distorted surface atomic structure in $Li_xRuO_2$, which is expected to modify the reactivity of the catalyst surface[44,50].

The free energies of the four elementary steps in OER ($* + 2H_2O \rightarrow OH^* \rightarrow O^* \rightarrow OOH^* \rightarrow O_2$) for $RuO_2$ and $Li_xRuO_2$ were calculated through DFT calculations, to uncover the role of the surface structure distortion. As (110) surface is the most stable surface of rutile $RuO_2$, two slab models of (110) surfaces were built for $RuO_2$ and $Li_{0.5}RuO_2$ (Supplementary Fig. 16). The Ru atom with a coordination number of 5 was considered as the active site[47]. The results (Fig. 3d) show that the

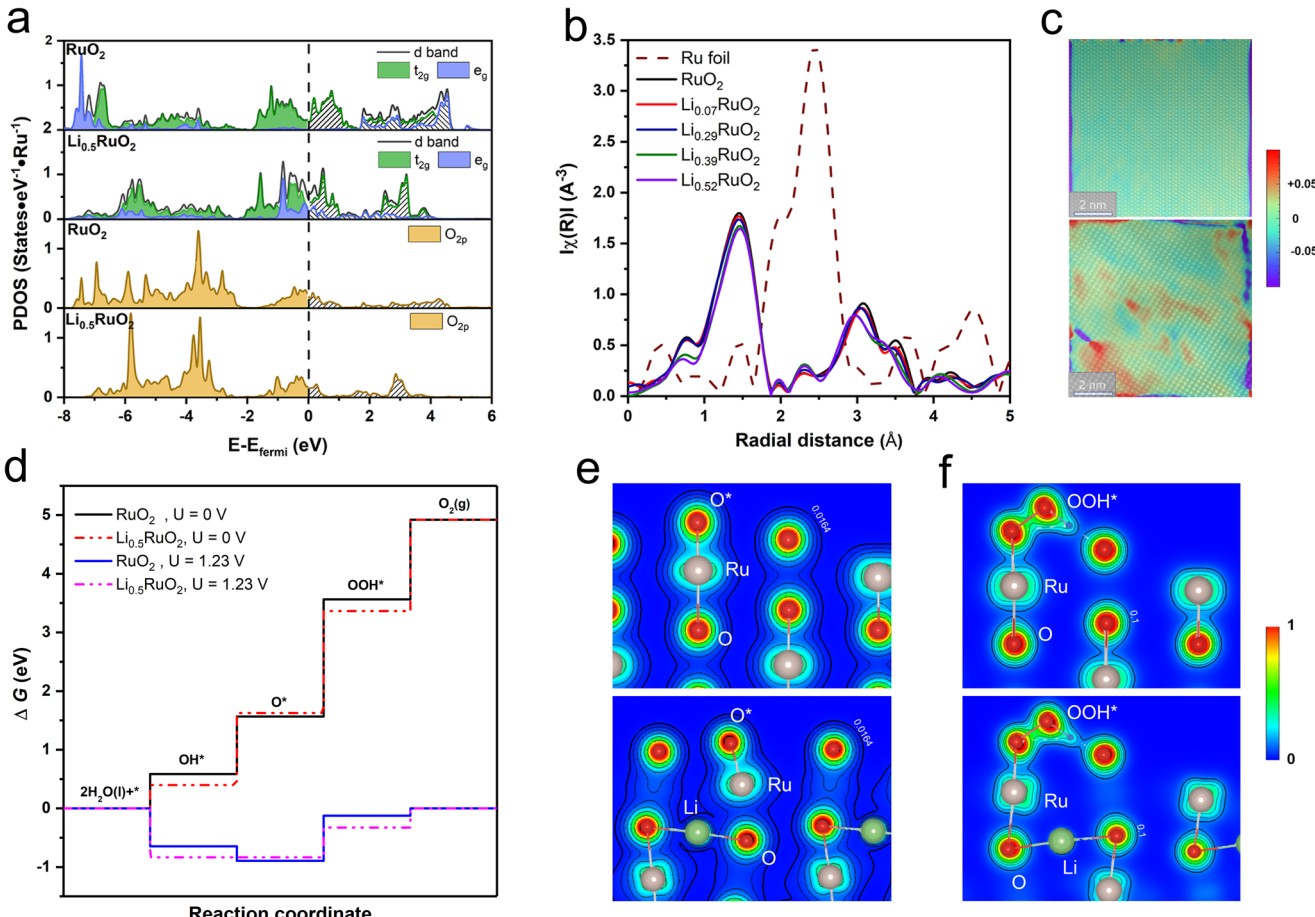

**Fig. 3 OER mechanism analysis. a** PDOS of the RuO$_2$ and Li$_{0.5}$RuO$_2$. **b** Fourier-transformed Ru K-edge extended X-ray absorption fine structure (EXAFS) spectra. **c** Lattice strain ($\varepsilon_{xx}$) measured from geometric phase analysis (GPA) of atomic-resolution HAADF − STEM images (Fig. 1f) for RuO$_2$ (up) and for Li$_{0.56}$RuO$_2$ (down). **d** Calculated OER free-energy diagrams for RuO$_2$ and Li$_{0.5}$RuO$_2$. **e** The charge density distribution of the O* absorbed on the (110) surface of RuO$_2$ (Up) and Li$_{0.5}$RuO$_2$ (down). The outermost black curve corresponds to the charge density of 0.0164 e$^-$/Bohr$^3$. **f** The charge density distribution of the OOH* absorbed on the (110) surface of RuO$_2$ (up) and Li$_{0.5}$RuO$_2$ (down). The outermost black curve corresponds to the charge density of 0.1 e$^-$/Bohr$^3$.

rate-determining step in the four-electron process for both RuO$_2$ and Li$_{0.5}$RuO$_2$ is the formation of OOH*, thence the absolute value Z ($\Delta$G(OOH*)-$\Delta$G(O*)) can be used to evaluate the OER catalytic activity. The Li$_{0.5}$RuO$_2$ model shows a Z value of ~1.74 eV, which is lower than that of RuO$_2$ (~2 eV). Subsequently, the energy consumption for the conversion from O* to OOH* is reduced at the surface of Li$_{0.5}$RuO$_2$. The decrease of Z in Li$_{0.5}$RuO$_2$ can be attributed to the decreased adsorption energy of O* and increased adsorption energy of OOH* (stabilization of OOH*) at the Li$_{0.5}$RuO$_2$ surface. Figure 3e, f compares the charge density distribution of the O* and OOH* absorbed on the (110) surface of RuO$_2$ and Li$_{0.5}$RuO$_2$. Interestingly, the obvious overlap of the electron cloud of the H atom of OOH* and the dangling O atom of Li$_{0.5}$RuO$_2$ is observed (Fig. 3f). The charge density in the center of the "bond" formed by the H atom of OOH* and the dangling O atom is calculated to be 0.091 and 0.132 e$^{-1}$ Bohr$^3$ for RuO$_2$ and Li$_{0.5}$RuO$_2$, respectively. Therefore, the dangling O atoms on the distorted surface are activated as a proton acceptor by lithium intercalation[3]. The H atom in OOH* can be more firmly bonded to the dangling O atom to stabilize the OOH*, resulting in a considerable improvement in the catalytic activity[3]. In sum, the enhanced activity of the Li$_x$RuO$_2$ is partially attributed to the modulated d-band structure of Ru, and more importantly is attributed to the lattice strain-induced activation of the dangling O atom as the proton acceptor. Therefore, the

OOH* vs. OH* scaling relation is broken and better activity is achieved. It is also worth noting that future efforts directed toward the ideal OER activity may focus on optimizing the free energy of every OER step to approach the equilibrium potential of 1.23 eV[51].

**Origin of the enhanced stability**. The prominent drawback of RuO$_2$ in acidic media is its poor stability, which is mainly due to the dissolution of high-valence Ru and oxidation of the lattice oxygen as a result of Ru–O covalency during the OER process[52]. Thus, it is necessary to decrease the valence state of Ru and suppress the participation of lattice oxygen. It is found that intercalation of lithium yields a Ru valence state of less than +4 and a decreased Ru–O covalency, as corroborated by the negative shift of the absorption edge position in the normalized Ru K-edge X-ray absorption near-edge structure (XANES) spectra for Li$_x$RuO$_2$ relative to that of RuO$_2$ (Fig. 4a)[17,19,53]. Figure 3b reveals that the bond length of Ru–O was slightly increased with the increase in lithium concentration. The evolution of the interatomic distances is consistent with the DFT calculations and indicates the languishing interaction of Ru–O[53], which may suppress the oxidation of lattice oxygen in OER[17,19,44,49,17,19,48,54]. From the O K-edge soft XAS (sXAS) as shown in Fig. 4b, the two peaks A$_1$ and A$_2$ represent the

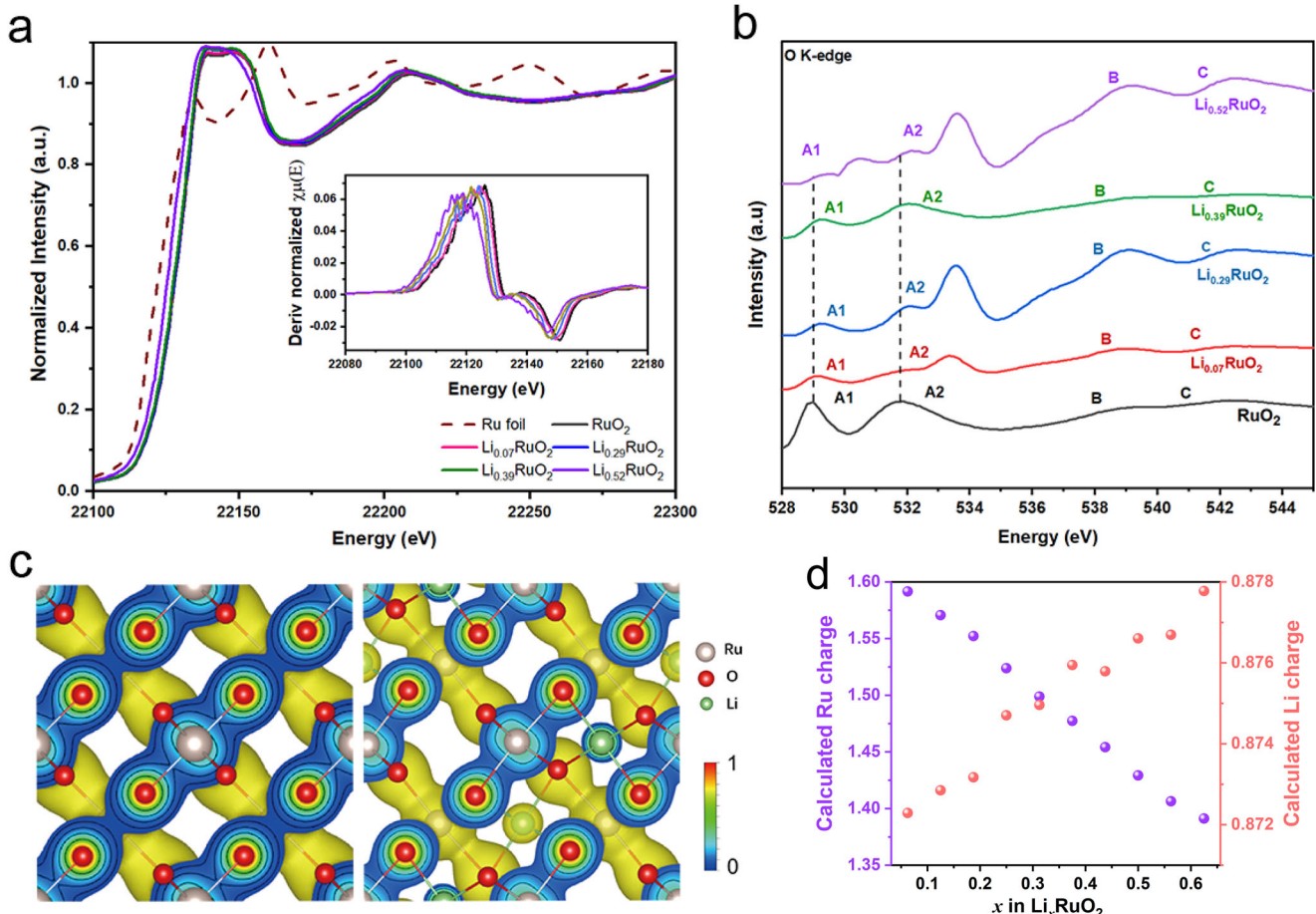

**Fig. 4 Electronic structure. a** Normalized Ru K-edge X-ray absorption near-edge structure (XANES) spectra. Inset: the first derivatives of the Ru K-edge XANES spectra of $RuO_2$ and $Li_xRuO_2$. **b** O K-edge soft XAS of $Li_xRuO_2$ and $RuO_2$. **c** Charge density distribution at the (110) crystal plane of $Li_xRuO_2$, with $n = 0$ (left) and 16 (right). **d** Ru and Li Bader charge.

excitations of the O 1 s core electrons into hybridized states of O $2p$ – Ru $4d$ $t_{2g}$ and O $2p$ – Ru $4d$ $e_g$ states[55]. The $A_1$ and $A_2$ peaks were clearly observed moving towards higher energy regions due to lithium intercalation, indicating the lowered covalency of Ru–O bond[56] as well as the reduced $Ru^4$, which is in good agreement with Ru K near-edge absorption results and the PDOS analyses (Fig. 3a)[55].

Figure 4c shows the charge density distribution at the (110) crystal plane of $Li_xRuO_2$ with $x = 0$ and 0.5, respectively. The Bader charges of Ru and Li are positive, indicating that the lithium atoms are electron donors (Fig. 4d). The donated electrons of a lithium atom increase slightly when increasing lithium concentration, while the donated electrons of Ru decrease gradually, indicating the decrease of the valence state of Ru cations. The donation of electron from Li to O indicates the formation of Li–O bond, and the bond strength is expected to be strengthened with the increase of lithium concentration. Therefore, the strong interaction in these Ru–O–Li local structure (Supplementary Fig. 17) may further suppress the lattice oxygen involvement during OER, thus improving the stability of the $Li_xRuO_2$[48]. Overall, on one hand, the lithium intercalation decreases the valence state of Ru, which enhances the resistance of Ru to dissolution in acidic solution. On the other hand, the lithium intercalation decreases the covalency of Ru–O bond and forms Ru–O–Li local structure, which suppresses the participation of lattice oxygen during OER.

In summary, the OER performance of $RuO_2$ was significantly improved by lithium intercalation, and reaches the best when the

nominal lithium concentration $x$ is 0.52 in $Li_xRuO_2$ solid solution phase. In particular, the $Li_{0.52}RuO_2$ possesses an ultralow overpotential of 156 mV for delivering a current density of 10 mA $cm^{-2}$ in 0.5 M $H_2SO_4$, with greatly enhanced durability. The excellent OER performance of $Li_xRuO_2$ is attributed to the dual function of lithium intercalation, i.e., modification of the electronic structure and tuning of the inherent lattice strain of $RuO_2$. The lithium donates electrons so that the valence state of Ru decreases, and interaction of Li–O increases. Meanwhile, the Ru−O $4d$ − $2p$ hybridization is weakened and the Ru–O covalency is decreased. Therefore, the participation of lattice oxygen and dissolution of Ru is suppressed during OER, enhancing the stability. On the other hand, the lithium intercalation modulates the $e_g$ occupancy of Ru $d$-band electrons to be closer to unity. Further, the inherent lattice strain results in the surface structural distortion, which activates the dangling O atom near the Ru active site as the proton acceptor, achieving stabilized OOH* and dramatically improved OER activity. This work proposes a creative strategy to simultaneously tune the electronic structure and lattice strain to design highly active and stable acidic OER catalysts for potential practical applications.

## Methods

**Sample preparation**. $Li_xRuO_2$ was prepared by electrochemical lithium intercalation. First, a working electrode was prepared by mixing $RuO_2$, carbon nanotubes (CNT), and PVDF (polyvinylidene fluoride) homogenously in n-methylpyrrolidone (NMP) with a weight ratio of 8:1:1, followed by coating the slurry on Cu foil and drying in an oven at 110 °C for 12 h. The working electrode was used to assemble CR2032 coin cells with lithium foil as the counter-electrode and 1 M solution of $LiPF_6$

in a mixture of ethylene carbonate (EC) and diethyl carbonate (DEC) (1:1 = v/v) as the electrolyte, in an argon-filled glovebox. The lithium intercalation into $RuO_2$ was achieved by discharging the cell at a constant current density of 0.05 C (1 C = 201.03 mA g$^{-1}$), while the content of lithium intercalated was controlled by the discharge time. After discharge, the cell was disassembled and the $RuO_2$ working electrode was washed using NMP several times to remove the PVDF and electrolyte, followed by drying at 60 °C to obtain $Li_xRuO_2$ powders.

**Characterization**. TEM images were collected on a JEOL JEM-1230 transmission electron microscope working at an operating voltage of 100 kV. HAADF-STEM photographs were collected on FEI Titan Themis Cube G2 high-resolution transmission electron microscope with 300 kV accelerating voltage. SEM images were recorded by Hitachi SU8230 microscope with 2 kV operating voltage. Ex situ XRD patterns of the powder samples of $Li_xRuO_2$ were measured on a Rigaku D/Max 2500 VB2 + /PC X-ray powder diffractometer by using Cu $K_\alpha$ radiation ($\lambda$ = 0.154 nm). *Operando* XRD measurements were performed on the same diffractometer using a self-designed in situ cell whose discharge-charge cycle was controlled by an electrochemical workstation. XPS measurements were executed at Thermo Scientific ESCALAB 250X with Al light source, and all binding energies were calibrated to the peak of C 1 s lied in 284.8 eV. XAS spectra at the K-edge of Ru were collected in transmission mode at beamline BL14W1 of 18KeV synchrotron radiation source at the SSRF, China. Soft XAS spectra of O K-edge were executed at beamline station BL12B in National Synchrotron Radiation Laboratory (NSRL), China, operated at 800 MeV with a maximum current of 300 mA. Neutron powder diffraction measurements were performed on the general-purpose powder diffractometer (GPPD) at the China Spallation Neutron Source (CSNS) in China.

**Electrochemical measurements**. Electrochemical measurements of $RuO_2$ and $Li_xRuO_2$ were performed in 0.5 M $H_2SO_4$ electrolyte with a standard three-electrode configuration controlled by an electrochemical workstation at room temperature. Oxygen gas was injected in the 0.5 M $H_2SO_4$ electrolyte for 10 min to ensure that the electrolyte is saturated with oxygen before electrochemical measurements. A catalyst-coated glassy carbon (GC) electrode (Diameter: 5 mm), Ag/AgCl electrode, and carbon rod were used as the working, reference and counter electrodes, respectively. In a typical scenario, 4 mg of $Li_xRuO_2$ powder was added to a mixed solution containing 200 μL ethanol and 200 μL Nafion aqueous solution (5 vol.%, ethanol as solvent), and dispersed by ultrasonication for 15 min to form a homogeneous black ink. The electrodes of $Li_xRuO_2$ were prepared by scribbling the ink on the GC electrode. The mass loading of Ru on each electrode is the same ~0.637 mg cm$^{-2}$ for all the samples. Linear sweep voltammetry (LSV) curves were conducted with a typical voltage range of 1.0–1.6 V vs. RHE and a scan rate of 10 mV/s. iR-compensation was not performed. Chronopotentiometric measurements were performed on a constant current of 10 mA cm$^{-2}$. Cyclic voltammetry (CV) measurements were conducted in the non-Faradaic region with different scan rates (5, 10, 20, 30, 40, and 50 mV s$^{-1}$). The electrochemically active surface areas (ECSA) were estimated from the electrochemical double-layer capacitance ($C_{DL}$) of the catalytic surface. The $C_{DL}$ was determined by plotting the $\Delta J/2$ ($\Delta J = J_a - J_c$, where $J_a$ is the anodic current and $J_c$ is the cathodic current at the middle voltage) against the scan rate, where the slope is equal to $C_{DL}$. The specific capacitance $C_s$ = 0.035 mF cm$^{-2}$ is used, and the ECSA is calculated according to ECSA = $C_{DL}/C_s$.

**DFT calculations**. The DFT calculations were performed using the Vienna ab initio Simulation Package (VASP)[57,58]. Perdew, Burke, and Ernzerhof (PBE) functional of generalized gradient approximation (GGA)[59] with projector augmented wave (PAW)[60] was applied to describe the electronic structures of materials. The plane-wave-basis kinetic energy cutoff was set to 450 eV. For the calculation of Li insertion, The Brillouin zones are sampled using Gamma-centered k-mesh of 5 × 5 × 5. For the calculations of the OER process, Van der Waals interaction is considered using the zero damping D3 method. The vacuum layers are set to ~15 Å to decouple the interaction between periodic images. The Brillouin zones are sampled using Gamma-centered k-mesh of 3 × 3 × 1. The slab models are built with 2 × 4 × 2 supercell and two bottom layers are fixed in the geometry optimization. The rest atomic layers and adsorbates are free to relax until the net force per atom is less than 0.02 eV/Å. The gas-phase $H_2$ and $H_2O$ molecules are optimized in a box of dimensions 15 × 15 × 15 Å with Gamma point sampling of the Brillouin zone. The adsorption energy ($E_{ad}$) is calculated by

$$E_{ad} = E_{total} - E_{slab} - E_{adsorbate},$$

where $E_{total}$ refers to the total energy of the optimized structure with the adsorbates absorbed on the slab surface, $E_{slab}$ refers to the energy of the clean slab, and $E_{adsorbate}$ refers to the energy of the adsorbate (O*, OH*, and OOH*) in vacuum.

## Data availability
The data that support the findings of this study are available within the article and its Supplementary Information files. All other relevant data supporting the findings of this study are available from the corresponding authors upon reasonable request. Source data are provided with this paper.

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

## Acknowledgements

This work was supported by research grants from the Guangdong Basic and Applied Basic Research Foundation (No. 2020A1515110798, No. 2022A1515012349), and Shenzhen Science and Technology Program (Grant No. RCBS20200714114920129). K.K.L. acknowledges the support by the Harbin Institute of Technology, Shenzhen. The authors thank the support from the general-purpose powder diffractometer (GPPD) at the China Spallation Neutron Source.

## Author contributions

T.-Y.Z. and K.L. managed the project and conceived the storyline of the paper. T.-Y.Z., K.L., X.-Y.Z., and H.-J.Q. guided the research. T.Y. performed the catalytic performance measurements, S.D. and L.H. conducted the NPD measurements and analyses, and Y.Q. conducted all the other experiments. D.L. participated in the in situ XRD measurements. Q.Z. performed the fitting of the XRD patterns using machine learning. Z.J. and D.Z. participated in the experiments. Y.-B.H., L.H., and F.K. contributed to the discussions of the project. X.-Y.Z. carried out the first-principles calculations and data analysis. H.-J.Q. managed the catalytic performance tests and analysis. K.L., T.-Y.Z., Y.Q., X.-Y.Z., and H.-J.Q. wrote the paper.

## Competing interests

The authors declare no competing interests.
