## [Peer Review File · Nature Communications]

RuO₂ electronic structure and lattice strain dual engineering for enhanced acidic oxygen evolution reaction performanceREVIEWER COMMENTS

Reviewer #1 (Remarks to the Author):

Manuscript NCOMMS-21-45806 studies the effect to Li doping of RuO₂ for the OER in acid electrolyte. This is indeed a hot topic in the literature and many interesting research works have appeared in the literature in the last years. In this work, Qin et al. report that Li doping to RuO₂ result in more active and durable OER catalyst. RuO₂ doping with several elements has been previously demonstrated to improve RuO₂ catalytic performance of the OER, and the doping of Ru mixed oxides with monovalent alkaline cations has been also demonstrated successful for increasing OER activity and performance. In this work, the authors chose to study the effect of Li, and synthesized a series of Li_xRuO₂ samples with different Li loadings by electrochemical methods. This is an interesting approach since controlling Li incorporation into oxides is not an easy task, neither it is the proper characterization of the Li-doped samples.

A weakness of this work relates to the characterization of Li cations in the sample, including concentration and location. I understand that this is a difficult task, but some of the conclusions of this work are based on the reported concentration and location of Li in the sample. For instance, it is stated that Li⁺ intercalates into the interstice of the RuO₂ lattice. I wonder how or to what extent this image represents the actual incorporation of Li cations. Can the authors provide experimental evidences to pinpoint the precise location of Li cations in Li_xRuO₂? Perhaps neutron powder diffraction would be helpful for this matter. The observed shifting of the diffraction lines to lower 2θ values has been ascribed to Li⁺ incorporation into the RuO₂ lattice. This is a plausible explanation if Li cations actually replace Ru cations in the lattice, since the ionic radii of Li⁺ cations in Oh position is actually larger than that of Ru⁴⁺ in Oh positions (0.76 vs 0.62 Å respectively). I wonder if this trend also applies for Li⁺ incorporated into lattice interstices. Would the incorporation of Li⁺ in interstitial positions result in a contraction of the cell?

Also, the actual content of Li into the samples is questionable. In S2 the authors offer a rough estimation of Li content based upon the discharge current. However, this method is not a direct measurement of Li content in the samples and some conclusions based on the reported figures should be taken carefully. For instance, Figure 4a shows a shifting of the Ru K-edge to lower energies for all Li containing samples with respect to RuO₂ standard. This is somehow expected since introducing Li⁺ will result in less oxidized Ru atoms. However, a careful observation of the figure reveals that position the Ru K-edge is similar for all Li doped samples, which is almost identical to that of RuO₂ (how representative is the very small shifting observed in the figures? Perhaps the first or second derivatives can shed light on this matter). The only real difference is for the sample containing the highest Li content, which shows a visible shifting. To some extent, this is also observed in the XPS spectra. The shifting of the Ru 3d_{5/2} peak is only observed in the spectrum of the sample with the highest loading of Li. How do the spectra of the Li 1s core-level region look like? Does the main peak shift in parallel to the Ru shifting observed in the Ru 3d_{5/2} core level region? Coming to that, can the authors provide the Ru/Li surface atomic ratios obtained from XPS?

Li⁺ doping increases the OER activity and durability of Li_xRuO₂, and the authors ascribe it to a combination of electronic and geometric effects caused by Li intercalation. This result is based on the shifting of the polarization curves with the potential (E₁₀) shifting to less positive values with the Li doping as shown in Figure 2a. Does this figure depict the first cycle or what cycle is that? Can the authors show several cycles (say from 1st to 10th or 20th cycle? In addition, does the current recorded at low potentials account to O₂ evolution? What is the actual loading of Ru on each electrode (does it change with Li content? Also, how does the presence of Li, especially in the high Li-loaded sample, affect the fraction of Ru atoms exposed at the catalyst surface? Although the determination of the ECSA for this kind of oxides may be tricky, it would be convenient to report area specific activities in order to further understand the origin of the higher OER activity.

Finally, in view of recent reports (see for instance Current Opinion in Electrochemistry 2018, 8 :110–117) , I wonder how important is to break the *OOH vs *OH scaling relation is the only valid strategy to designing more active OER catalyst.

Reviewer #2 (Remarks to the Author):

This paper describes an extensive investigation into the effect of lithium insertion on the electrocatalytic properties of RuO₂. The paper reports an extensive set of experimental measurements that encompass structural investigation, spectroscopy and electrochemistry, further supported by some DFT calculations. Overall this leads to a consistent set of conclusions regarding the enhanced activity and superior acid stability of Li_xRuO₂, which is of importance for practical applications in water splitting electrochemical devices.

I believe that the paper is of sufficient quality and importance to warrant publication in Nature Communications, but I have two suggestions of how the paper might be improved.

Firstly, there is already an extensive literature around the insertion of lithium in RuO₂ for battery applications, and this should be mentioned, with the inclusion of some relevant citations. Earlier literature on lithium insertion in RuO₂ includes:

<https://doi.org/10.1002/adfm.200304406>

<https://iopscience.iop.org/article/10.1149/1.1939147>

<https://pubs.rsc.org/en/content/articlelanding/2020/ta/d0ta06428a>

<https://doi.org/10.1021/jp5123536>

Secondly, the authors discuss the stability of the electrocatalyst in terms of dissolution of RuO₂ (pages 9-10 and Figure 2e). This is important for real applications, and although a comparison with the literature on dissolution of RuO₂ is made, the authors should report their own data from RuO₂ to make a proper assessment of the improvement on lithium insertion. The dissolution will depend on particle size and shape as well as the condition of the experiment and so it is important to include data measured under the same conditions.

Reviewer #3 (Remarks to the Author):

Regarding the theoretical work by Qin et al., the most significant results are summarized in Figure 3. However, I do not believe that their DFT calculations adequately support the experimental observations. The Li-intercalated RuO₂ structure shows an almost identical free energy diagram compared to the pristine RuO₂ system, where the free energy of converting *O to *OOH changes from ~1.9 eV to ~1.7 eV. The following step, in fact, of converting *OOH to O₂(g), becomes more uphill by a similar amount, and may even be slightly larger than 1.7 eV. These numbers are not in agreement with their experimental data in Figure 2. Based on the computational work, I do not recommend publication in Nature Communications.

Responses to Reviewers' comments

We highly appreciate the valuable comments from the editor and the reviewers. We have taken all the comments and revised the manuscript accordingly. We truly believe that the quality of the manuscript has been greatly improved after this revision. In the following, we give our responses item-by-item after each of the comments and list the changes made in the revised version.

Reviewer #1

General Comments. Manuscript NCOMMS-21-45806 studies the effect to Li doping of RuO₂ for the OER in acid electrolyte. This is indeed a hot topic in the literature and many interesting research works have appeared in the literature in the last years. In this work, Qin et al. report that Li doping to RuO₂ result in more active and durable OER catalyst. RuO₂ doping with several elements has been previously demonstrated to improve RuO₂ catalytic performance of the OER, and the doping of Ru mixed oxides with monovalent alkaline cations has been also demonstrated successful for increasing OER activity and performance. In this work, the authors chose to study the effect of Li, and synthesized a series of Li_xRuO₂ samples with different Li loadings by electrochemical methods. This is an interesting approach since controlling Li incorporation into oxides is not an easy task, neither it is the proper characterization of the Li-doped samples.

Response. We sincerely appreciate your valuable comments.

Comment 1. A weakness of this work relates to the characterization of Li cations in the sample, including concentration and location. I understand that this is a difficult task, but some of the conclusions of this work are based on the reported concentration and location of Li in the sample. For instance, it is stated that Li⁺ intercalates into the interstice of the RuO₂ lattice. I wonder how or to what extent this image represents the actual incorporation of Li cations. Can the authors provide experimental evidences to pinpoint the precise location of Li cations in Li_xRuO₂? Perhaps neutron power diffraction would be helpful for this matter. The observed shifting of the diffraction lines to lower 2theta values has been ascribed to Li⁺ incorporation into the RuO₂ lattice. This is a plausible explanation if Li cations actually replace Ru cations in the lattice, since the ionic radii of Li⁺ cations in Oh

position is actually larger than that of Ru^{4+} in Oh positions (0.76 vs 0.62 Å respectively). I wonder if this trend also applies for Li^+ incorporated into lattice interstices. Would the incorporation of Li^+ in interstitial positions result in a contraction of the cell?

Response 1. Thank you so much for the valuable and insightful comment. We took the comment and performed neutron powder diffraction (NPD) as well as density functional theory (DFT) calculations to further investigate the location of Li cations in Li_xRuO_2 . Both the NPD analyses and DFT calculations of Li_xRuO_2 demonstrate that the Li ions should intercalate into the interstitial sites of RuO_2 rather than replace the Ru cations. The incorporation of Li^+ in the interstitial positions results in the expansion of the cell, which is demonstrated by the XRD results (Supplementary Table 1) as well as the DFT calculations. The revised text and associated figures are copied below.

Revised manuscript, Page 6:

“Neutron powder diffraction (NPD) analyses (Supplementary Fig. 4) and DFT calculations (Supplementary Fig. 5-6, Fig. 1c) indicate that the lithium ions intercalate into the octahedral interstice formed by six adjacent O atoms rather than replace the Ru cations, and the RuO_2 lattice is expanded, which is in line with the XRD results.”

Revised supplementary information, Page 5-8:

“Supplementary Text 1:

Determination of the location of Li ions in Li_xRuO_2

Neutron powder diffraction (NPD) patterns of pristine RuO_2 and $\text{Li}_{0.52}\text{RuO}_2$ were collected on the general purpose powder diffractometer (GPPD) at the China Spallation Neutron Source, and crystal structures were determined by the Rietveld refinement method with the General Structure Analysis System (GSAS)¹. The results are shown in Supplementary Fig. 4. Obviously, both the pristine RuO_2 and $\text{Li}_{0.52}\text{RuO}_2$ maintain the rutile RuO_2 structure. Assuming that the Li atoms occupy the Ru sites in $\text{Li}_{0.52}\text{RuO}_2$, the refinement indicates that the occupancy of Li is only $\sim 0.008(3)$, which means Li atom hardly enters the Ru sites. The NPD patterns of $\text{Li}_{0.1}\text{Ru}_{0.9}\text{O}_2$ and $\text{Li}_{0.34}\text{Ru}_{0.66}\text{O}_2$, in which

10% and 34% Ru cations were replaced by Li cations, were simulated using GSAS program. It is found that the substitution of Ru by Li decreases the intensity of (110) and (211) peaks but has ignorable influence on the (111) and (210) peaks. Comparison of the experimental NPD data of $\text{Li}_{0.52}\text{RuO}_2$ with the simulated patterns of $\text{Li}_{0.1}\text{Ru}_{0.9}\text{O}_2$ and $\text{Li}_{0.34}\text{Ru}_{0.66}\text{O}_2$ further indicates that the Li atoms didn't replace the Ru cations. Thereby, we conclude that the Li^+ intercalates into the interstice of the RuO_2 lattice. In addition, the background intensity of the NPD pattern of RuO_2 is obviously enhanced after lithium insertion, implying the lattice distortion/strain induced by lithium insertion.

Supplementary Figure 4 | Rietveld refinement of (a) RuO_2 NPD data and (b) $\text{Li}_{0.52}\text{RuO}_2$ NPD data. Please note that the peaks originating from the CNT and by-products of the electrochemical process were treated as background and subtracted during the refinement. (c) Comparison of the experimental NPD data of $\text{Li}_{0.52}\text{RuO}_2$ and simulated NPD patterns of $\text{Li}_{0.1}\text{Ru}_{0.9}\text{O}_2$ and $\text{Li}_{0.34}\text{Ru}_{0.66}\text{O}_2$. In $\text{Li}_{0.1}\text{Ru}_{0.9}\text{O}_2$ and $\text{Li}_{0.34}\text{Ru}_{0.66}\text{O}_2$, 10% and 34% Ru sites are occupied by Li atoms, respectively.

DFT calculations were further performed to investigate the location of Li ions in Li_xRuO_2 from the energy point of view. The energetics of inserting Li into interstitial sites of RuO_2 and replacing the Ru atoms of RuO_2 were calculated, respectively.

Scenario I: Li is inserted into the O-octahedron interstice.

Li atoms were inserted gradually into the O-octahedron interstice in the $\text{Ru}_{16}\text{O}_{32}$ supercell. For the selection of O-octahedrons for Li insertion, we enumerated all the possible combination and chose the combination with the lowest energy for further calculation. We then performed structural relaxation of models with Li inserted. The energy cost for Li insertion is calculated by

$$E_{\text{insertion}} = E_{\text{Ru}_{16}\text{O}_{32}+n\text{Li}} - E_{\text{Ru}_{16}\text{O}_{32}} - nE_{\text{Li}},$$

where n is the number of Li atoms inserted, $E_{\text{Ru}_{16}\text{O}_{32}+n\text{Li}}$ is the energy of RuO_2 supercell with Li atoms inserted, $E_{\text{Ru}_{16}\text{O}_{32}}$ is the energy of the RuO_2 supercell and E_{Li} is the energy of a Li atom in vacuum. As more Li atoms are inserted, $E_{\text{insertion}}$ becomes more negative, indicating Li insertion is exothermic and spontaneous. $E_{\text{insertion}}$ is plotted as a function of n in Supplementary Fig. 5.

Supplementary Figure 5 | $E_{\text{insertion}}$ as a function of n .

Scenario II: Li replaces Ru in RuO_2 .

The substitutional energy for replacing Ru by Li in RuO_2 superlattice is calculated as

$$E_{\text{substitution}} = E_{\text{Li}_n\text{Ru}_{16-n}\text{O}_{32}} - E_{\text{Ru}_{16}\text{O}_{32}} + nE_{\text{Ru}} - nE_{\text{Li}},$$

where n is the number of Li atoms to replace Ru atoms, $E_{\text{Li}_n\text{Ru}_{16-n}\text{O}_{32}}$ is the energy of RuO_2 supercell with n Ru atoms replaced by n Li, $E_{\text{Ru}_{16}\text{O}_{32}}$ is the energy of the RuO_2 supercell, E_{Ru} and E_{Li} are the energies of a Ru and Li atom in vacuum, respectively. $E_{\text{substitution}}$ is always positive and as n increases, $E_{\text{substitution}}$ becomes more positive, indicating that it is energetically unfavorable for Li to replace Ru in RuO_2 superlattice. $E_{\text{substitution}}$ is plotted as a function of n in Supplementary Fig. 6.

Supplementary Figure 6 / $E_{\text{substitution}}$ as a function of n .

From the energy point of view, we can conclude that Li ions tend to insert into the O-octahedron interstice rather than replace the Ru atoms in RuO_2 .

Comment 2. Also, the actual content of Li into the samples is questionable. In S2 the authors offer a rough estimation of Li content based upon the discharge current. However, this method is not a direct measurement of Li content in the samples and some conclusions based on the reported figures should be taken carefully. For instance, Figure 4a shows a shifting of the Ru K-edge to lower energies for all Li containing samples with respect to RuO_2 standard. This is somehow expected since introducing Li^+ will result in less oxidized Ru atoms. However, a careful observation of the figure reveals that position the Ru K-edge is similar for all Li doped samples, which is almost identical to that of RuO_2 (how representative is the very small shifting observed in the figures? Perhaps the first or second derivatives can shed light on this matter). The only real difference is for the sample containing the highest Li content, which shows a visible shifting. To some extent, this is

also observed in the XPS spectra. The shifting of the Ru $3d_{5/2}$ peak is only observed in the spectrum of the sample with the highest loading of Li.

Response 2. Many thanks for the valuable and useful comment. Characterization of the Li concentration in Li_xRuO_2 is truly a difficult task. We have tried various methods to measure the Li concentration, which include inductively coupled plasma mass spectrometry (ICP-MS), Electron Energy Loss Spectroscopy (EELS), and X-ray Photoelectron Spectroscopy (XPS). However, we cannot obtain consistent results through these methods because of various reasons (*e.g.*, the Li_xRuO_2 can hardly be dissolved in acid solutions for ICP-MS, the EELS signal is not sensitive enough to Ru, inevitable contamination of samples by carbon-related impurities, and small sensitivity factors of Li and Ru for XPS), which means we cannot get the actual content of Li in Li_xRuO_2 . Therefore, we use “nominal Li concentration” to denote the Li concentration estimated based upon the discharge current in our revised manuscript. Considering that this method is commonly used in battery research to determine the lithium concentration in an electrode material, we think it is a relatively reliable method to estimate the lithium concentration in Li_xRuO_2 . In addition, the Li_xRuO_2 is a solid solution phase when $x < 1$ with no phase transition (*J. Phys. Chem. C* 2015, 119, 9705), and the exact lithium concentration of $\text{Li}_{0.52}\text{RuO}_2$ won't be greater than 0.8 from the potential profile analysis. Therefore, it is reasonable to assume that the physical/chemical/electronic properties of Li_xRuO_2 change monotonously with x in the range of 0 ~ 1. Under this condition, even though the exact lithium concentration is unknown, we think our experimental and calculational analyses based on the nominal lithium concentration are reliable to illustrate the effect of lithium insertion on the OER performance of RuO_2 .

Revised manuscript, Page 6:

“Fig. 1e presents the ex situ XRD patterns of the pristine RuO_2 and the Li_xRuO_2 after electrochemical lithium intercalation under a current density of 10 mA g^{-1} for 2h, 9h, 12h, and 16h, corresponding to the nominal lithium concentrations of $x = 0.07, 0.29, 0.39,$ and 0.52 (Details for the estimation of the nominal lithium concentration can be found in Supplementary Fig. 2).”

To highlight the shifting of the Ru K-edge, we followed the comment and plotted the first

derivatives of the Ru K-edge XANES spectra, and the results are shown in Figure 4a in the revised version. It is found that the Ru K-edge is gradually shifted to low energy direction with increase of the Li concentration, and the shift is quite visible for all the Li doped samples. As for the inapparent shifting of the Ru $3d_{5/2}$ XPS peak, there should be some reasons but we are sorry that we cannot explain it. Therefore, we deleted the XPS results in the revised manuscript as the XANES results are sufficient to prove the reduced valence state of Ru due to the lithium insertion.

Revised manuscript, Page 15:

Fig. 4 Electronic structure. **a** Normalized Ru K-edge X-ray absorption near edge structure (XANES) spectra. Inset: The first derivatives of the Ru K-edge XANES spectra of RuO_2 and Li_xRuO_2 . **b** O K-edge soft XAS of Li_xRuO_2 and RuO_2 . **c** Charge density distribution at the (110) crystal plane of $\text{Li}_x\text{RuO}_{0.5}$, with $x = 0$ (Left) and 0.5 (Right). **d** Ru and Li Bader charge.

Comment 3. How do the spectra of the Li 1s core-level region look like? Does the main peak shift in parallel to the Ru shifting observed in the Ru $3d_{5/2}$ core level region? Coming to that, can the authors provide the Ru/Li surface atomic ratios obtained from XPS?

Response 3. Many thanks for the comment. We followed the comment and measured the Li 1s XPS of Li_xRuO_2 , which are shown in Supplementary Fig. 9a in the revised version. The results indicate that the main peak of the Li 1s XPS shows no shift of the peak position. We tried to determine the Ru/Li surface atomic ratios by XPS but failed. The possible reason is that the sensitivity factors of Li and Ru for XPS are very small, so that the intensities of the XPS peaks and the signal-to-noise ratios of the XPS survey spectra are very weak. Thus, we cannot obtain a precise Ru/Li atomic ratio by XPS.

Revised supplementary information, Page 12:

Supplementary Figure 9 | (a) The high-resolution Li 1s XPS of Li_xRuO_2 . The HAADF-STEM image (b) and Li K-edge STEM-EELS (c) of the $\text{Li}_{0.52}\text{RuO}_2$.

Comment 4. Li^+ doping increases the OER activity and durability of Li_xRuO_2 , and the authors ascribe it to a combination of electronic and geometric effects caused by Li intercalation. This result is based on the shifting of the polarization curves with the potential (E10) shifting to less positive values with the Li doping as shown in Figure 2a. Does this figure depict the first cycle or what cycle is that? Can the authors show several cycles (say from 1st to 10th or 20th cycle)? In addition, does the current recorded at low

potentials account to O₂ evolution? What is the actual loading of Ru on each electrode (does it change with Li content? Also, how does the presence of Li, especially in the high Li-loaded sample, affect the fraction of Ru atoms exposed at the catalyst surface? Although the determination of the ECSA for this kind of oxides may be tricky, it would be convenient to report area specific activities in order to further understand the origin of the higher OER activity.

Response 4. Thank you very much for the valuable comment. Figure 2a depicts the polarization curves of the 3rd cycle of each sample. We took the comment and added the polarization curves of Li_{0.52}RuO₂ from the 1st to the 7th cycle, which are shown in Supplementary Fig. 11. It can be found that the polarization curves show almost no change in the initial 7 cycles.

The current at low potentials does account to the O₂ evolution. To clarify this issue, we optically observed the bubble evolution on the electrode surface when the Li_{0.52}RuO₂ is under chronovoltometric tests at different voltages of 1.2, 1.3, 1.4 and 1.5 V, respectively. As shown in Supplementary Fig. 11, we found that when the voltage was 1.3 V, gas bubbles started to appear on the working electrode, and became more and more with increase of the voltage.

Revised manuscript, Page 9:

“Supplementary Fig. 11 shows that the O₂ generation starts at around 1.3 V, and the polarization curve shows almost no change in the initial 7 cycles. Here, the overpotential for reaching a current density of 10 mA cm⁻² (η_{10}) of the 3rd cycle is used for activity comparison.”

Revised supplementary information, Page 14:

Supplementary Figure 11 | (a) Optical images of the $\text{Li}_{0.52}\text{RuO}_2$ electrode surface when the electrode is under chronovoltmetric tests. (b) Polarization curves of $\text{Li}_{0.52}\text{RuO}_2$ at different cycles.

The fabricated loading of Ru on each electrode is the same $\sim 0.637 \text{ mg cm}^{-2}$ for all the samples. We performed DFT calculations to investigate the influence of Li insertion on the fraction of Ru atoms exposed at the (110) surface of $\text{Li}_{0.5}\text{RuO}_2$. The Li atoms causes the expansion of the lattice in comparison with the virgin lattice. Although the Li insertion induced strain varies the fraction of Ru atoms exposed at the (110) surface, the variation is much smaller than the observed influence of Li on the OER activity.

As mentioned in the comment, the determination of ECSA (electrochemically active surface areas) would be convenient to further understand the origin of the higher OER

activity. We took the comment and measured the ECSA, and reported the ECSA-based LSVs (Supplementary Fig. 13), which indicate that the higher OER activity of Li_xRuO_2 is not attributed to the varied ECSA, and the Li insertion plays an important role in enhancing the intrinsic activity.

Revised manuscript, Page 9:

“We further estimated the electrochemically active surface area (ECSA) of RuO_2 and Li_xRuO_2 , and plotted the LSVs with respect to the ECSA (Supplementary Fig. 12-13), which indicate that the higher OER activity of Li_xRuO_2 is not attributed to the varied ECSA, and the Li insertion plays an important role in enhancing the intrinsic activity.”

Revised manuscript, Page 19:

“Cyclic voltammetry (CV) measurements were conducted in non-Faradaic region with different scan rates (5, 10, 20, 30, 40 and 50 mV s^{-1}). The electrochemically active surface areas (ECSA) were estimated from the electrochemical double-layer capacitance (C_{DL}) of the catalytic surface. The C_{DL} was determined by plotting the $\Delta j/2$ ($\Delta j = j_a - j_c$, where j_a is the anodic current and j_c is the cathodic current at the middle voltage) against the scan rate, where the slope is equal to C_{DL} . The specific capacitance $C_s = 0.035 \text{ mF cm}^{-2}$ is used, and the ECSA is calculated according to $\text{ECSA} = C_{DL}/C_s$.”

Revised supplementary information, Page 15-16:

Supplementary Figure 12 | Cyclic voltammograms of the (a) pristine RuO_2 and the RuO_2 after lithiation (b) $\text{Li}_{0.07}\text{RuO}_2$, (c) $\text{Li}_{0.29}\text{RuO}_2$, (d) $\text{Li}_{0.39}\text{RuO}_2$, and (e) $\text{Li}_{0.52}\text{RuO}_2$, respectively, collected at scan rates of 5, 10, 20, 30, 40, and 50 mV s^{-1} in 0.5 M H_2SO_4 solution. (f) Capacitive current density plotted against scan rate and linear fitting for the evaluation of C_{DL} .

Supplementary Figure 13 | ECSA-based LSVs of the pristine RuO_2 and Li_xRuO_2 .

Comment 5. Finally, in view of recent reports (see for instance *Current Opinion in Electrochemistry* 2018, 8 :110–117), I wonder how important is to break the $*\text{OOH}$ vs $*\text{OH}$ scaling relation is the only valid strategy to designing more active OER catalyst.

Response 5. Thank you very much for the suggestive comment. As we know, for OER process with the AEM mechanism, there are four elemental steps and each step involves a concerted proton–electron transfer reaction. In principle, any of these four steps could be the potential-determining step of the OER process, and an ideal catalyst requires all these steps with the same reaction free energies (*i.e.*, 1.23 eV). However, because the adsorption energies of the OER intermediates (HO^* , HOO^* , and O^*) are linearly correlated, particularly, the difference between the adsorption energies of HO^* and HOO^* are ~ 3.2 eV, the theoretical overpotential cannot go beyond 0.37 eV. Therefore, breaking the OOH^* vs OH^* scaling relation is considered to be very important for the design of more active OER catalyst.

The above conclusion is also established under the consideration that the first step (the adsorption of OH^- on the surface O vacancy site) and the fourth step (the deprotonation of HOO^*) rarely act as the potential-determining step in most OER catalysts. Obviously, in

the cases of the fourth step being the potential-determining step, breaking the OOH^* vs OH^* scaling relation may not lead to the improved activity. The paper mentioned in the comment (**Current Opinion in Electrochemistry 2018, 8 :110–117**) reviewed the literature data and found that a number of catalysts breaking the OOH^* vs OH^* scaling relation factually possess high overpotentials, suggesting that breaking the OOH^* vs OH^* scaling relation is a necessary yet insufficient condition to optimize OER electrocatalysts. Alternatively, the authors proposed a new descriptor, the electrochemical-step symmetry index (ESSI), which was defined as $(\Delta G_i - 1.23 \text{ eV})/n$, where ΔG_i is the free energy of the steps and only applies to those larger than 1.23 eV, n is the number of the steps whose free energy is larger than 1.23 eV. They showed that the ESSI values can explain why high OER activity may be predicted on materials that do not break scaling relations, and why low OER activity may as well be predicted on materials that break those. Thus, it can be concluded that future efforts directed toward the design of enhanced OER catalysts may focus not only on breaking the HOO^* and HO^* scaling relation, but also on optimizing ESSI, which also means optimizing the free energy of every OER step to approach the equilibrium potential of 1.23 eV.

Revised manuscript, Page 14:

“Therefore, the OOH^ vs. OH^* scaling relation is broken and better activity is achieved. It is also worth noting that future efforts directed toward the ideal OER activity may focus on optimizing the free energy of every OER step to approach the equilibrium potential of 1.23 eV⁵¹.”*

Reviewer #2

General Comments. This paper describes an extensive investigation into the effect of lithium insertion on the electrocatalytic properties of RuO_2 . The paper reports an extensive set of experimental measurements that encompass structural investigation, spectroscopy and electrochemistry, further supported by some DFT calculations. Overall this leads to a consistent set of conclusions regarding the enhanced activity and superior acid stability of Li_xRuO_2 , which is of importance for practical applications in water splitting

electrochemical devices.

I believe that the paper is of sufficient quality and importance to warrant publication in Nature Communications, but I have two suggestions of how the paper might be improved.

Response. We sincerely appreciate your valuable comments.

Comment 1. Firstly, there is already an extensive literature around the insertion of lithium in RuO₂ for battery applications, and this should be mentioned, with the inclusion of some relevant citations. Earlier literature on lithium insertion in RuO₂ includes:

<https://doi.org/10.1002/adfm.200304406>

<https://iopscience.iop.org/article/10.1149/1.1939147>

<https://pubs.rsc.org/en/content/articlelanding/2020/ta/d0ta06428a>

<https://doi.org/10.1021/jp5123536>

Response 1. Many thanks for the suggestive comment. We took the comment and cited these papers in our revised manuscript. These papers do provide deep insights of the lithiation mechanism of RuO₂, which are consistent to our findings. Revised text is copied below.

Revised manuscript, Page 4:

“Various studies have shown that RuO₂ can be inserted with Li ions for battery applications, and a solid solution phase forms before a Li:Ru=1:1 ratio is reached^{35, 36, 37, 38}.”

Comment 2. Secondly, the authors discuss the stability of the electrocatalyst in terms of dissolution of RuO₂ (pages 9-10 and Figure 2e). This is important for real applications, and although a comparison with the literature on dissolution of RuO₂ is made, the authors should report their own data from RuO₂ to make a proper assessment of the improvement on lithium insertion. The dissolution will depend on particle size and shape as well as the condition of the experiment and so it is important to include data measured under the same

conditions.

Response 2. Thanks very much for the valuable comment. We took the comment and measured the dissolution of the pristine RuO₂ electrode under the same conditions to that of the Li_{0.52}RuO₂ electrode (chronopotentiometry test at 10 mA cm⁻²). In addition, we demonstrated that the lithium intercalation shows no influence on the morphology and size of the pristine RuO₂ particles, as shown in Supplementary Fig. 8. The results showed that the percentage of Ru dissolved from pristine RuO₂ during the chronopotentiometry test was factually smaller than that of Li_{0.52}RuO₂, as shown in Figure 2e in the revised manuscript. This is reasonable because the activity of RuO₂ is quite poor (Figure 2a) and decreases dramatically in less than 20 h (Figure 2d).

As mentioned in the comment, the dissolution will depend on the condition of the experiment. Thus, it is important to compare the dissolution at the same condition. Following the comment, we indicated clearly the testing conditions for different catalysts when comparing with the literatures in the revised manuscript. The testing condition is either the same to our test (*J. Am. Chem. Soc.* 2021, 143, 6482) or more moderate than our test (*Angew. Chem. Int. Ed.* 2021, 60, 18821). The revised text and figures are copied below.

Revised manuscript, Page 10:

“The percentage of Ru dissolved from pristine RuO₂ and Li_{0.52}RuO₂ during the chronopotentiometry tests at 10 mA cm⁻² was measured and shown in Fig. 2e. For pristine RuO₂, the dissolution percentage of Ru is very low because of its low activity and poor stability. For Li_{0.52}RuO₂, in the 1st hour of the OER test, the dissolution percentage of Ru is around 0.9%. After 24 h, the dissolution percentage of Ru is increased slightly to 1.8%, and then plateaued. Even after 48 h, the dissolution percentage of Ru remained very low at 1.9% which is much lower than those reported for amorphous/crystalline hetero-phase RuO₂ (in 0.1 M HClO₄)⁴⁷ and SrRuIr oxide (in 0.5 M H₂SO₄)⁴⁸ during chronopotentiometry test at 10 mA cm⁻², indicating good corrosion resistance of Li_{0.52}RuO₂ in acidic condition.”

Revised manuscript, Page 8:

Fig. 2 OER performance in 0.5 M H_2SO_4 solution. **a** Polarization curves. RHE: Reversible Hydrogen Electrode. **b** Overpotentials (η_{10}) of RuO_2 and Li_xRuO_2 at 10 mA cm^{-2} . **c** Tafel plots. **d** Chronopotentiometry curve of $Li_{0.52}RuO_2$ and RuO_2 at a current density of 10 mA cm^{-2} . **e** Percentage of Ru dissolved from RuO_2 and $Li_{0.52}RuO_2$ after electrocatalysis for different reaction times. **f** Comparison of the overpotential required to achieve a 10 mA cm^{-2} cathodic current density and chronopotentiometry durability at 10 mA cm^{-2} in acidic media for various RuO_2 -based electrocatalysts^{14, 17, 23, 25, 40, 41, 42, 43}.

Reviewer #3

Comments. Regarding the theoretical work by Qin et al., the most significant results are summarized in Figure 3. However, I do not believe that their DFT calculations adequately support the experimental observations. The Li-intercalated RuO_2 structure shows an almost identical free energy diagram compared to the pristine RuO_2 system, where the free energy of converting $*O$ to $*OOH$ changes from $\sim 1.9\text{ eV}$ to $\sim 1.7\text{ eV}$. The following step, in fact, of converting $*OOH$ to $O_2(g)$, becomes more uphill by a similar amount, and may even be slightly larger than 1.7 eV . These numbers are not in agreement with their experimental data in Figure 2. Based on the computational work, I do not recommend publication in Nature Communications.

Response. We thank the reviewer very much for pointing out the shortcomings of our calculation work, which is very helpful to improve our manuscript. We re-examined our calculation details and found the following fault. The energies of the slab models were overrated so that the energy required for oxygen desorption, which was the last step of the four-electron process, was largely overrated. This is because, in our previous calculations, we used a slab model with a $2 \times 2 \times 2$ superlattice, which is too small to manifest the surface reconstruction associated with the position changes of surface oxygen atoms. Therefore, we adopted a larger slab model with a $2 \times 4 \times 2$ superlattice which is expanded along the y direction in our revised calculations, to simulate the surface reconstruction. Obviously, it is found that the surfaces of the slab models underwent reconstruction, with the positions of the O atoms changed significantly. Under this condition, we re-performed the DFT calculations of the adsorption energies of the adsorbates and the Gibbs free energy changes during the OER process. The results showed that the free energy of converting O^* to OOH^* is ~ 2 eV for RuO_2 and ~ 1.74 eV for $Li_{0.5}RuO_2$, revealing a significant reduction of the overpotential by Li insertion. In addition, the free energy of converting OOH^* to $O_2(g)$ is changed from 1.35 eV for RuO_2 to 1.55 eV for $Li_{0.5}RuO_2$.

Indeed, the overpotentials predicted by the DFT calculations are still not exactly the same to the experimental data, although the DFT calculations demonstrate the obvious reduction of overpotential by Li insertion. In fact, the calculated energy barriers for converting O^* to OOH^* of RuO_2 and $Li_{0.5}RuO_2$ are very close to those calculated for some excellent RuO_2 -based catalyst reported in literatures (*Nat. Commun.* 2020, 11: 5368; *Nat. Commun.* 2019, 10: 162; *Angew. Chem. Int. Ed.* 2021, 60: 18821). In these literatures, the calculated energy barriers for converting O^* to OOH^* of RuO_2 are 2.02 eV, 2.02 eV, and 2.22 eV, which are reduced to 1.76 eV ($W_{0.2}Er_{0.1}Ru_{0.7}O_{2-\delta}$), 1.87 eV ($Cr_{0.6}Ru_{0.4}O_2$), and 1.74 eV (amorphous/crystalline hetero-phase RuO_2), respectively. Their experimental overpotentials (η_{10}) are reduced from ~ 300 mV, 297 mV, and 279 mV to 168 mV, 178 mV, and 205 mV, respectively. Therefore, the DFT calculations are intended to understand the trends of the activity variation when the RuO_2 is inserted by Li. The possible reasons for the difference between the experimental overpotentials and calculated ones are the slab model is still too small and the neighboring intermediates around the active sites are not considered. In the real scenario of the OER process, there should be a large number of different intermediates absorbed on those active sites at the same time, and the energetics

of the elementary processes at one single active site will be influenced significantly by the neighboring intermediates, which will contribute to enhancing the activity (*Nat. Commun.* **2020**, *11*: 5368). However, DFT calculations based on the above considerations will be extremely difficult and time-consuming. We sincerely appreciate your comments for improving our manuscript, and we think the revised calculations now provided a rational explanation for the experimental data. The revised text and associated figures are copied below.

Revised manuscript, Page 11, 13-14:

“As (110) surface is the most stable surface of rutile RuO₂, two slab models of (110) surfaces were built for RuO₂ and Li_{0.5}RuO₂ (Supplementary Fig. 16). The Ru atom with a coordination number of 5 was considered as the active site⁴⁷. The results (Fig. 3d) show that the rate-determining step in the four-electron process for both RuO₂ and Li_{0.5}RuO₂ is the formation of OOH, thence the absolute value Z ($\Delta G(\text{OOH}^*) - \Delta G(\text{O}^*)$) can be used to evaluate the OER catalytic activity. The Li_{0.5}RuO₂ model shows a Z value of ~1.74 eV, which is lower than that of RuO₂ (~2 eV). Subsequently, the energy consumption for the conversion from O* to OOH* is reduced at the surface of Li_{0.5}RuO₂. The decrease of Z in Li_{0.5}RuO₂ can be attributed to the decreased adsorption energy of O* and increased adsorption energy of OOH* (stabilization of OOH*) at the Li_{0.5}RuO₂ surface. Figures 3e-f compares the charge density distribution of the O* and OOH* absorbed on the (110) surface of RuO₂ and Li_{0.5}RuO₂. Interestingly, obvious overlap of electron cloud of the H atom of OOH* and the dangling O atom of Li_{0.5}RuO₂ is observed (Fig. 3f). The charge density in the center of the “bond” formed by the H atom of OOH* and the dangling O atom is calculated to be 0.091 and 0.132 e⁻¹ Bohr³ for RuO₂ and Li_{0.5}RuO₂, respectively. Therefore, the dangling O atoms on the distorted surface are activated as a proton acceptor by lithium intercalation³. The H atom in OOH* can be more firmly bonded to the dangling O atom to stabilize the OOH*, resulting in a considerable improvement in the catalytic activity³.”*

Fig. 3 OER mechanism analysis. **a** PDOS of the RuO_2 and $\text{Li}_{0.5}\text{RuO}_2$. **b** Fourier-transformed Ru K-edge extended X-ray absorption fine structure (EXAFS) spectra. **c** Lattice strain (ϵ_{xx}) measured from geometric phase analysis (GPA) of atomic-resolution HAADF-STEM images (Fig. 1f) for RuO_2 (Up) and for $\text{Li}_{0.5}\text{RuO}_2$ (Down). **d** Calculated OER free-energy diagrams for RuO_2 and $\text{Li}_{0.5}\text{RuO}_2$. **e** The charge density distribution of the O^* adsorbed on the (110) surface of RuO_2 (Up) and $\text{Li}_{0.5}\text{RuO}_2$ (Down). The outermost black curve corresponds to the charge density of $0.0164 e/\text{Bohr}^3$. **f** The charge density distribution of the OOH^* adsorbed on the (110) surface of RuO_2 (Up) and $\text{Li}_{0.5}\text{RuO}_2$ (Down). The outermost black curve corresponds to the charge density of $0.1 e/\text{Bohr}^3$.

Revised manuscript, Page 20-21:

“The slab models are built with $2 \times 4 \times 2$ supercell and two bottom layers are fixed in the geometry optimization. The rest atomic layers and adsorbates are free to relax until the net force per atom is less than $0.02 \text{ eV}/\text{\AA}$. The gas-phase H_2 and H_2O molecules are optimized in a box of dimensions $15 \text{ \AA} \times 15 \text{ \AA} \times 15 \text{ \AA}$ with Gamma point sampling of the Brillouin-zone. The adsorption energy (E_{ad}) is calculated by

$$E_{ad} = E_{total} - E_{slab} - E_{adsorbate},$$

where E_{total} refers to the total energy of the optimized structure with the adsorbates absorbed on the slab surface, E_{slab} refers to the energy of the clean slab, and $E_{adsorbate}$ refers to the energy of the adsorbate (O^* , OH^* , and OOH^*) in vacuum.”

Revised supplementary information, Page 19-20:

Supplementary Figure 16 | DFT-optimized structures of RuO_2 and $Li_{0.5}RuO_2$ and adsorption energies (E_a) of the oxo-intermediates.

Supplementary Figure 17 | The charge density distribution of (a) pristine RuO_2 , (b) $\text{Li}_{0.5}\text{RuO}_2$, and the OH^* adsorbed on the (110) surface of (c) RuO_2 and (d) $\text{Li}_{0.5}\text{RuO}_2$. The outermost black curve corresponds to the charge density of $0.0164 e^-/\text{Bohr}^3$.

REVIEWERS' COMMENTS

Reviewer #1 (Remarks to the Author):

The authors have addressed all of my remarks, in particular they conducted further experiments (PND, DFT) for determining the location of Li. Although the location and actual Li loading of the samples is not completely resolved, I recognize that this is a very difficult task and that the authors have conducted a thorough research.

Before acceptance, I have a minor comment. Figure 2a inset: it seems that currents of ca 2-4 mA/cm² are recorded already at potentials below 1.23 V for some of the catalysts. This is strange, at such low potentials no OER currents should be recorded. How were these curves obtained? Usually full CVs rather than LSV are recorded, and the averaged current (averaging the forward and reverse scans) are represented. Can the authors show the complete CVs (forward and reverse scans?)

Reviewer #3 (Remarks to the Author):

In response to my previous concerns about their DFT calculations not supporting experimental results, the authors have carried out additional calculations with a larger unit cell, and examined in more detail the charge density map and projected density of states to analyze their DFT-calculated Gibbs free energies. The authors have also revised their statement to speak about trends in the adsorption energies. In light of these new calculations, which are in better agreement with previous work published in Nature Communications and Angewandte Chemie, I believe this work is acceptable for publication in Nature Communications.

Responses to Reviewers' comments

We highly appreciate the valuable and positive comments from the editor and the reviewers. We have taken the comments and revised the manuscript accordingly. In the following, we give our responses item-by-item after each of the comments.

Reviewer #1

Comments. The authors have addressed all of my remarks, in particular they conducted further experiments (PND, DFT) for determining the location of Li. Although the location and actual Li loading of the samples is not completely resolved, I recognize that this is a very difficult task and that the authors have conducted a thorough research.

Before acceptance, I have a minor comment. Figure 2a inset: it seems that currents of ca 2-4 mA/cm² are recorded already at potentials below 1.23 V for some of the catalysts. This is strange, at such low potentials no OER currents should be recorded. How were these curves obtained? Usually full CVs rather than LSV are recorded, and the averaged current (averaging the forward and reverse scans) are represented. Can the authors show the complete CVs (forward and reverse scans?)

Responses. We sincerely appreciate your valuable comments. These curves are Linear Sweep Voltammetry (LSV) curves, which were obtained by linearly sweeping the potential from 1.0 V to 1.6 V vs. RHE at a scan rate of 10 mV/s. Following the comment, we recorded the full CV of Li_{0.52}RuO₂ in the potential range of 1.0 V – 1.6 V vs. RHE at a scan rate of 10 mV/s, which is shown as followed and added in the revised Supplementary Information (Supplementary Figure 11c). The CV curve indicates that the currents that recorded at potentials below 1.23 V originate from the charge of double layer.

Supplementary Figure 11c. Cyclic voltammetry curves of $\text{Li}_{0.52}\text{RuO}_2$ in the potential range of 1.0 V – 1.6 V vs. RHE.

Reviewer #3

Comments. In response to my previous concerns about their DFT calculations not supporting experimental results, the authors have carried out additional calculations with a larger unit cell, and examined in more detail the charge density map and projected density of states to analyze their DFT-calculated Gibbs free energies. The authors have also revised their statement to speak about trends in the adsorption energies. In light of these new calculations, which are in better agreement with previous work published in Nature Communications and Angewandte Chemie, I believe this work is acceptable for publication in Nature Communications.

Responses. We sincerely appreciate your valuable comments.